# Subcutaneous Application of a Gelatin/Hyaluronic Acid Hydrogel Induces the Production of Skin Extracellular Matrix

**DOI:** 10.3390/polym16050573

**Published:** 2024-02-20

**Authors:** Katia Jarquín-Yáñez, Miguel Ángel Herrera-Enríquez, Diego Ivan Benítez-Barrera, Francisco M. Sánchez-Arévalo, Jorge Alejandro Benítez-Martínez, Gabriela Piñón-Zárate, Beatriz Hernández-Téllez, Diana M. Aguilar Sandoval, Andrés E. Castell-Rodríguez

**Affiliations:** 1Facultad de Medicina, National Autonomous University of Mexico, Mexico City 04510, Mexico; katys12@hotmail.com (K.J.-Y.); mikeh@unam.mx (M.Á.H.-E.); gabrielapinon@unam.mx (G.P.-Z.); bhernandezt@hotmail.com (B.H.-T.); dianitamichel@ciencias.unam.mx (D.M.A.S.); 2Materials Research Institute, National Autonomous University of Mexico, Mexico City 04510, Mexico; diego-ivan00@hotmail.com (D.I.B.-B.); fsanchez@iim.unam.mx (F.M.S.-A.); jbenitez1393@gmail.com (J.A.B.-M.)

**Keywords:** hydrogels, EDC, skin fillers, biocompatible materials, tissue engineering

## Abstract

The development of injectable hydrogels with natural biopolymers such as gelatin (Ge) and hyaluronic acid (Ha) is widely performed due to their biocompatibility and biodegradability. The combination of both polymers crosslinked with N-Ethyl-N′-(3-dimethyl aminopropyl) carbodiimide hydrochloride (EDC) can be used as an innovative dermal filler that stimulates fibroblast activity and increases skin elasticity and tightness. Thus, crosslinked Ge/Ha hydrogels with different concentrations of EDC were administered subcutaneously to test their efficacy in young and old rats. At higher EDC concentrations, the viscosity decreases while the particle size of the hydrogels increases. At all concentrations of EDC, amino and carboxyl groups are present. The histological analysis shows an acute inflammatory response, which disappears seven days after application. At one and three months post-treatment, no remains of the hydrogels are found, and the number of fibroblasts increases in all groups in comparison with the control. In addition, the elastic modulus of the skin increases after three months of treatment. Because EDC-crosslinked Ge/Ha hydrogels are biocompatible and induce increased skin tension, fibroblast proliferation, and de novo extracellular matrix production, we propose their use as a treatment to attenuate wrinkles and expression lines.

## 1. Introduction

Facial fillers have been widely used to reduce wrinkles, fill in expression lines, and improve face esthetics [1]. Additionally, they have addressed other skin problems, such as volume loss, skin laxity, cellulite, and swollen stretch marks, offering a more youthful skin appearance through minimally invasive procedures [1,2]. Dermal fillers add volume; provide immediate results with lower cost, risk, and discomfort than conventional surgery; and allow for a prompt recovery [1]. However, these benefits are variable and relate to the biological response of each patient together with other individual factors, such as age, immune status, previous treatments, and pathological medical history, which determine the effectiveness of the dermal fillers [2].

In these aesthetic interventions, soft tissue augmentation is achieved using various biomaterials: dermal fillers can be fluids, biological fragments, or suspensions of particles and microspheres [3]. They are classified as temporary (or biodegradable), permanent (or non-biodegradable), or a combination of both. Among the most commonly used biodegradable fillers, hyaluronic acid (Ha), collagen (bovine, porcine, and human), poly-L-lactic acid, calcium hydroxyapatite, and dextran pearls in Ha are available as fillers. As for non-biodegradable fillers, polymethylmethacrylate microspheres with bovine collagen, polymethyl methacrylate microspheres suspended in carboxy gluconate gel, silicone, saturated hydrocarbons, polymethylmethacrylate silicone suspension, acrylic hydrogel particles suspended in Ha, polyacrylamide gel, polyvinyl microspheres suspended in polyacrylamide, e-polytetrafluoroethylene, and Gore-Tex are some of the available options in the market [1,3,4]. However, their degradation time is not optimal, and they are also used as fillers, but not as cell or tissue stimulators.

Hydrogels are crosslinked hydrophilic polymer chains with multiple biomedical and pharmaceutical applications as dermal fillers [5]. The type of gelation used in the synthesis of hydrogels distinguishes between two general categories: stable hydrogels, characterized by the presence of covalent bonds; and unstable hydrogels, which are reversible through the application of force or environmental changes and result from the presence of ionic bonds, hydrogen bonds, physical interactions, or molecular entanglements [5]. Hydrogels are characterized by their ability to absorb water at quantities higher than their dry weight and rapidly degrade and dissolve. The physicochemical properties of hydrogels depend on their composition, which is based on a wide variety of natural or synthetic polymers or a combination of both [6].

Ha is a polymer of natural origin that has been used in manufacturing hydrogels; the latter are used in clinical processes due to their biocompatibility, degradability, high water retention capacity, safety, and high cell interaction [7]. It has been used in the form of hydrogel, dermal filler, intradermal injection, scaffold, cream, film, foam, and gel for the treatment of different types of diseases [8]. Ha stimulates the synthesis of extracellular matrix metalloproteases and modulates the immunological responses. The biological properties of Ha depend on its molecular weight, with the high-molecular weight Ha (>1 × 10^6^ Da) inducing antiangiogenic and immunosuppressive effects and promoting the repair of skin wounds in the absence of keloid or hypertrophic healing [9].

Combination with natural polymers, such as gelatin (Ge), has been applied to improve Ha hydrogels’ degradation time and rheological properties [9,10]. This process can be carried out through physical crosslinking with UV radiation [10] or through chemical processes that mediate the esterification of the hydroxyl groups (-OH), the acetylation and deacetylation of the acetyl groups (NHCOCH3), or the amidation and oxidation of the carboxyl groups (-COOH) [9,10,11,12,13,14]. In this context, N-(3-Dimethylaminopropyl)-N′-ethylcarbodiimide (EDC) is a chemical crosslinker that forms covalent bonds between the carboxylic acid and the amino groups, leading to the formation of amides that functionalize the Ha hydrogel linked to another natural polymer, such as Ge [11,15]. Ge is a hydrolyzed form of collagen; it has low immunogenicity and contains arginine–glycine–aspartate peptide regions with high affinity for αV β3 integrins, which are critical molecules in cell adhesion [15]. No filler material can be considered to possess all of the specifications appropriate for all medical indications or for treating all kinds of patients. Therefore, a mixture of biomaterials can increase the necessary properties that a skin filler should have. In this sense, molecules of the extracellular matrix, such as the combination of Ge and Ha, may be used as a skin filler that can mediate the migration and stimulation of fibroblasts, relevant characteristics in skin fillers. Although the combination of Ge/Ha has previously been proposed as a dermal filler [16], the EDC has not been evaluated as a crosslinker to form a Ge/Ha hydrogel serving as a skin filler, nor has its efficacy been tested in the dermis of young and old rats. Moreover, it should be noted that no Ge/Ha dermal filler is currently used in medicine [17].

Thus, this study aimed to analyze the mechanical properties and histological characteristics of the skin after subcutaneous injection with different concentrations of EDC-crosslinked Ge/Ha composite hydrogel.

## 2. Materials and Methods

### 2.1. Synthesis of Ge/Ha Hydrogel Crosslinked with EDC at Different Concentrations

The hydrogel was manufactured from a homogenized solution of Ge type B obtained from bovine skin (Sigma-Aldrich, St. Louis, MO, USA), with a molecular weight (MW) of 20,000–25,000, and Ha obtained from *Streptococcus equi*, with a MW of 403.3 g/mol (Sigma-Aldrich, St. Louis, MO, USA) dissolved in distilled water at 8% and 2%, respectively, at a temperature of 50 °C. Next, 1 mL of the solution was placed in a conic tube (total of 20 tubes), cooled to room temperature for 1 h, and stored at −20 °C overnight. The solution was lyophilized for 10 h at a pressure of 0.02 hPa and a temperature of −110 °C in a lyophilizer (Labogene, ScanVac, Lillerød, Denmark), and it was subsequently crosslinked by immersion with EDC (Sigma-Aldrich, St. Louis, MO, USA) at a concentration of 6 mM, 30 mM, and 50 mM dissolved in 96% alcohol, according to Jarquín et al. 2016 [15]. The scaffolds were stored at −20 °C for 12 h. Subsequently, they were washed three times with distilled water to remove the excess EDC. The crosslinked scaffolds were lyophilized for 10 h under the aforementioned conditions and cut into small discs with a thickness of 2 mm and a diameter of 9 mm. Discs weighing 0.4 g were hydrated with 15 mL of PBS for 24 h and then subjected to mechanical pulverization for 30 s at 1000 rpm on six occasions using a porcelain homogenizer. The obtained hydrogel was sterilized with UV light for 30 min and stored at −4 °C until use.

### 2.2. Determination of Particle Size

To identify the size of the particles, 50 μL of each hydrogel were placed on microscopy slides and photographed under a Nikon Eclipse 80i microscope (Tokyo, Japan); the samples were also analyzed in a low-vacuum scanning electron microscope (SEM; LV-5600; JEOL, Peabody, MA, USA). The size distribution of the hydrogel microparticles was measured using a particle analyzer laser (LA-950V2, HORIBA, Kyoto, Japan) at 25.0 °C ± 0.5 °C using distilled water as the dispersion medium.

### 2.3. Determination of Viscosity

To measure the viscosity, 10 mL of each hydrogel was placed in a concentric cylinder in a DV2T Brookfield Ametek viscometer (Middleboro, MA, USA at 10 rpm with an SC4-16 spindle. A temperature of 25 °C was used in the experimental groups and 37 °C in the control group to avoid Ge polymerization.

### 2.4. Infrared Spectroscopy (IR)

All hydrogels were analyzed using an infrared spectrometer (NICOLET Nexus 670 FT-IR; ALT, San Diego, CA, USA) to determine their molecular composition. Infrared spectra were registered using the transmittance technique on potassium bromide pellets (Buck Scientific, Norwalk, CT, USA). All experiments were performed in triplicate.

### 2.5. Subcutaneous Application of Ge/Ha Hydrogel

For this study, 40 male Wistar rats were obtained from and housed at the Animal Facility of the Faculty of Medicine, UNAM. Animal care and treatments were approved by the Ethical and Research Commissions of the Faculty of Medicine (UNAM; FM/DI/125/2017). All animals were kept under pyrogen-free conditions and controlled light–dark cycles with ad libitum access to water and Purina chow pellets.

The hydrogel treatment was evaluated for its short-term effects (1, 2, 4, 7, and 14 days post-application) with juvenile rats (three months old; Ν = 3 per condition), while the hydrogel long-term effects (one and three months post-application) were analyzed in both juvenile (three months old; Ν = 3 per condition) and old rats (one year old; Ν = 3 per condition); all groups are summarized in Table 1. Before applying the hydrogels, the rats were anesthetized with 80 mg/kg of ketamine (PiSA Pharmaceutical, Mexico City, Mexico) and 10 mg/kg of xylazine (PiSA Pharmaceutical, Mexico City, Mexico) intramuscularly. Subsequently, under aseptic conditions, the skin of the dorsal region was depilated, and the area was disinfected with 10% povidone-iodine (Betadine^®^, Madrid, Spain).

In the case of short-term effects, all individuals received a 50 µL hydrogel dose (Figure 1A). For the long-term effects analysis, the hydrogel was distributed throughout the skin of the dorsal region of the rat by 20 inoculations of 50 microliters (Figure 1B).

### 2.6. Histological Evaluation of Ge/HA Hydrogels

To obtain the biological samples, the rats were euthanized with 120 mg/kg of sodium pentobarbital administered intraperitoneally as per the requirements of NOM-033-SAG/ZOO-2014. Afterward, a skin biopsy was obtained from the site of the hydrogel application (Table 1). The samples were fixed in 15 mL of 10% formalin for 24 h. Then, they were processed using conventional histological techniques and stained for the H&E and Herovici techniques.

To evaluate the migration of the neutrophils and macrophages at the application site of the hydrogel, a time course analysis was carried out at one, two, four, seven, and 14 days post-inoculation. Low-magnification images (40×) were obtained, and the presence or remnants of hydrogels and the inflammatory process were verified. Meanwhile, at 400×, in 10 histological fields of each experimental group, neutrophils and macrophages were counted in order to analyze the leucocytes migration to the hydrogel’s implantation site. In addition, to evaluate the long-term histological changes in the skin of young and old rats after the inoculation of the hydrogels, skin biopsies were collected after one- and three-months post treatment. Also, at 400× images of reticular dermis the number of fibroblasts per mm^2^ was quantified. Whereas to evaluate the new collagen fibers matrix synthesis the Herovici stain was performed. All stained slides were observed under a photonic microscope (Eclipse 80i, Nikon, Tokyo, Japan) and analyzed using the Image ProPlus 7.0 image analyzer (Media Cybernetics, Rockville, MD, USA).

### 2.7. Mechanical Properties of the Skin under Uniaxial Tension

The mechanical analysis of the skin was conducted under uniaxial tension. For this purpose, a custom-designed mechanical tester was used. The mechanical tester was composed of a stainless steel load frame coupled to a linear mechanical actuator (modelT-LA60A from Zaber Technologies, Vancouver, BC, Canada) serving as a linear motion source, providing up to 15 N of loading force and a controlled displacement in a range of 60 mm. In addition, a precision miniature load cell (model 31, from Honeywell Sensotec, Columbus, OH, USA) was used to measure the axial load during the tensile tests. The load cell and mechanical actuator were in the axis load of the load frame, coupled with 3D-printed grips that were specially fabricated to hold the skin samples firmly, avoiding any damage in the grip region during the tensile test. The data were acquired through an NI-cDAQ-9174 chassis and an NI- board (National Instruments, Austin, TX, USA). All signals were synchronized through a customized virtual instrument programmed in LabVIEW 2023 Q3 version, a software from National Instruments, Austin, TX, USA.

As for displacement and force, the stress vs. elongation ratio curves were obtained using a displacement rate of 0.16 mm/s. From these curves, the shear modulus of each sample was determined using a second-order Ogden model [18], as shown in Equation (1):(1)$$\sigma =\sum_{i=1}^{n=2}\mu_{i}*(\lambda^{\alpha_{i}}-\lambda^{{(-\alpha }_{i}/2)})$$
where *σ* is the stress, *λ* is the elongation or elongation ratio (*λ* = *ε* + 1), *μ* is the shear modulus ($\mu =\frac{1}{2}\sum_{i=1}^{n}\mu_{i}*\alpha_{i}$), and *α* is a parameter related to the strain invariants for incompressible materials according to the strain energy function proposed by Ogden [18]. Using data from the uniaxial tensile experiments and considering the Ogden model, the parameters $\mu_{i}$ and $\alpha_{i}$ can easily be estimated by non-linear fitting. Once the shear modulus was determined, the elastic modulus was calculated using the elastic constants relation E = 2 *μ* (1 + *ν*), assuming that *ν* = 1/2 for materials with rubber-like mechanical behavior.

### 2.8. Statistic Analysis

Each experimental condition (EDC crosslinker concentration used in the manufacture of the hydrogel) consisted of N = 3 for each of the experimental times.

To analyze the results, GraphPad Prism 9.3.1 software (San Diego, CA, USA) was used to perform descriptive statistics and, subsequently, for each experiment, a two-way analysis of variance (ANOVA) with a Sidak multiple comparison test was executed. *p* < 0.05 was considered significant in all experiments.

## 3. Results

### 3.1. Ge/Ha Particle Size Analysis

A mixture of Ge and Ha solutions was subjected to crosslinking and lyophilization to modulate the porosity of the resulting scaffold concerning their size and the number of pores per area. As a result, a porous polymeric scaffold of Ge/Ha was obtained. The physical and biological characterization of the scaffold has been previously published [15]. A representative image of the EDC-crosslinked scaffold under SEM is shown in Figure 2a,b. After the pulverization process, among the scaffolds, the particles were not homogeneous, as evidenced by the presence of fibers of varying thickness and size and by the trabeculae of the polymeric hydrogel (Figure 2c). When the size of the particles was analyzed, those corresponding to the 50 mM EDC-crosslinked hydrogels were larger than those that were crosslinked with EDCs of 6 mM and 30 mM (Figure 2d) (* *p* < 0.0001).

### 3.2. Ge/Ha Viscosity Analysis

The viscosity of the hydrogels obtained after the pulverization of Ge/Ha scaffolds crosslinked with different concentrations of EDC was analyzed. It is noteworthy that the viscosity of the Ge and Ha hydrogels (control groups) showed large differences, with the Ge hydrogels presenting a mean viscosity of 14.33 cP and the Ha hydrogels having a mean viscosity of 136,000 cP (*p* < 0.001). The hydrogels crosslinked with EDC at 6 mM (649.90 cP, *p* < 0.001) and 30 mM (62.63 cP, *p* < 0.001) showed much higher viscosity than that observed in the 50 mM EDC (12.10 cP, *p* < 0.001) and Ha (14.33 cP, *p* < 0.001) hydrogels. The viscosity of the Ge/Ha hydrogels gradually and significantly decreased with increasing EDC concentrations. None of the hydrogels showed a higher viscosity than the Ha hydrogel (Figure 3).

### 3.3. The Composition of Ge/Ha Hydrogels

The different concentrations of EDC used to manufacture the Ge/Ha scaffolds did not change their composition. The IR spectra revealed in the curves of all groups an amide group at the wavelength of 1545 cm^−1^. The peaks of all hydrogels corresponding to COOH (1614 cm^−1^), C=O (1650 cm^−1^), C-H (1080 cm^−1^), and C-C (1415 cm^−1^) were similar to the control group (non-crosslinked Ge/Ha; Figure 4). In addition, the curves of all experimental groups were superimposed but had different magnitudes that represented a change in the hydration of the hydrogels. At 6 mM EDC, the Ge/Ha hydrogel showed the highest hydration, while at 30 mM and 50 mM EDC, the hydrogels displayed overlapping curves, translating into lower water absorption.

### 3.4. Effect of the Application of Ge/Ha Hydrogels in the Dermis of Rats

The hydrogels were subcutaneously implanted in rats, and the histological changes were evaluated 1 and 14 days later to analyze the degradation and inflammatory response.

Regarding the inflammatory response, at one and two days post-injection, all experimental groups presented inflammatory cell infiltrates with a basophilic appearance invading the application site (Figure 5a). We noticed that no inflammatory infiltrates were detected in the connective tissue surrounding the application site. At four days post-application, the inflammatory infiltrates significantly decreased in all experimental groups, with the group of 6 mM EDC showing the largest decrease. By day seven post-application, no leukocytes were observed in the groups that were treated with 6 mM and 30 mM EDC hydrogels. However, leukocytes were still observed in the group that was treated with the 50 mM EDC-crosslinked hydrogels. Finally, 14 days after the application of the hydrogels, no inflammatory infiltrate was observed in any of the experimental groups (Figure 5a).

Figure 5b shows representative images of the inflammatory infiltrate induced by applying the hydrogels over time. Also, a temporal curve was constructed to establish the type of inflammatory infiltrate induced by the hydrogels (Figure 5c). At one and two days post-application, polymorphonuclear cells were the most predominant type of cell with a density of 12.4 × 10^3^ cells/mm^2^. In the case of mononuclear cells, the first two days after application, approximately 4–2 × 10^3^ cells/mm^2^ was found, and they reached their maximum levels on day four after the application of the hydrogels (6.7 × 10^3^ cells/mm^2^). Subsequently, the number of leukocytes gradually and continuously decreased until day 14 post-application.

### 3.5. Effect of Ge/Ha Hydrogels on the Dermal Extracellular Matrix

To evaluate the effect of hydrogel administration on the skin of juvenile and old rats, at one and three months post-injection, a histological analysis, the quantification of fibroblasts in the extracellular matrix, and the measurement of the percentage of synthesized collagen I and III were performed.

In the group of juvenile rats, at both one and three months post-injection, the fibroblast cell density increased significantly compared with the groups treated with Ha or PBS (control groups). Moreover, more fibroblasts were found in the dermis of all groups at three months after Ge/Ha treatment compared with one month of treatment (Figure 6a). In contrast, no significant changes in the number of fibroblasts were found in old rats between one and three months of treatment, regardless of the EDC concentration. Nonetheless, a significant increase in the number of fibroblasts was found in rats treated with Ge/Ha hydrogels compared with the Ha group (Figure 6b).

As for the presence of a recently synthesized extracellular matrix characterized by abundant type III collagen and immature collagen I fibers, the matrix was evaluated using cerulean blue with the Herovici staining technique. A statistically significant increase in the quantification parameters of the new matrix was found in both juvenile and old rats treated with Ge/Ha hydrogels compared with the control group. Notably, this increase was directly proportional to the concentration of EDC; the dermis of the rats treated with the 50 mM EDC-crosslinked hydrogels presented a newer matrix than those treated with the hydrogels crosslinked with 6 mM EDC. It is important to highlight that the degree of new matrix production was higher in the juvenile rats group than in the old rats group (Figure 6c).

### 3.6. Mechanical Behavior of the Rat Skin

The uniaxial test results revealed a non-linear mechanical behavior, which is evident at the beginning of the stress vs. elastic modulus curves (Figure 7). Figure 7a shows the mechanical behavior of the juvenile rats’ skin after one month of treatment. The obtained maximum elongation ratio ranged between 1.5 and 1.9, and the maximum stress values were between 1.4 MPa and 2.2 MPa. The average values of the elastic moduli were 0.30 ± 0.1 MPa, 0.045 ± 0.07 MPa, 0.031 ± 0.01 MPa, and 0.051 ± 0.02 MPa, corresponding to Ha, 6 mM, 30 mM, and 50 mM EDC, respectively.

Figure 7b depicts the mechanical response of the skin of juvenile rats after three months of treatment. Here, the stress vs. elongation ratio curves also presented a non-linear mechanical behavior. However, the internal fibers of the skin tended to align faster after three months of treatment compared with one month of treatment. The obtained maximum elongation ratio was between 1.35 and 1.5. The maximum stress values were between 1.2 MPa and 1.9 MPa. The applied treatment produced a material with higher elastic moduli (Figure 7b), reaching elastic moduli of 0.59 ± 0.5 MPa, 0.29 ± 0.3 MPa, 1.43 ± 0.8 MPa, and 1.62 ± 1 MPa, corresponding to Ha, 6 mM, 30 mM, and 50 mM EDC, respectively.

In the case of the old rats treated for one month, the elongation ratio ranged between 1.55 and 1.6, while rats treated for three months exhibited elongation ratios ranging between 1.4 and 1.55. The maximum stress values ranged between 0.5 MPa and 1.8 MPa for one month of treatment and between 1.45 MPa and 1.8 MPa for the rats with three months of treatment. Regarding one month post-treatment, the corresponding elastic moduli were 0.20 ± 0.06 MPa, 0.023 ± 0.09 MPa, 0.31 ± 0.2 MPa, and 0.27 ± 0.3 MPa, while for three months post-treatment, they were 0.20 ± 0.06 MPa, 0.26 ± 0.02 MPa, 0.37 ± 0.5 MPa, and 0.63 ± 0.4 MPa for Ha, 6 mM, 30 mM, and 50 mM EDC, respectively.

Considering these results, we observed that the average values were close to the order of the standard deviation, which is expected due to the nature of the biological tissue nature. However, we detected an increasing trend in the elastic modulus depending on the concentration of EDC. At three months post-application, the treatment was more effective in juvenile rats than in old rats. At the highest concentration of EDC (50 mM), the elastic modulus value of juvenile rats was threefold compared with that of old rats. Thus, Ge/Ha hydrogels in younger rats for longer periods offer more resistant skin under uniaxial loads.

## 4. Discussion

Nowadays, beauty standards have led to a growing demand for non-invasive dermatological procedures to rejuvenate aged skin. In the last 20 years, more and more laboratory and clinical trials have intended to develop techniques that minimize wrinkles and expression lines, impede UV light-induced damage, and restore the volume of the skin’s connective tissue [19,20,21].

On the other hand, it is well-documented that skin aging results from intrinsic and extrinsic factors that produce free radicals [20,21]. The increased production of free radicals induces the secretion of matrix metalloproteinases by fibroblasts, macrophages, and neutrophils residing in the dermis, which produces the fragmentation of collagen and elastic fibers [22]. The fragmentation of collagen and elastic fibers causes the detachment of fibroblasts, leading to a further increase in metalloproteinase secretion and a decrease in the synthesis of extracellular matrix molecules [23]. This results in a reduced volume and quantity of the extracellular matrix, making it more vulnerable to further enzymatic degradation, which explains the appearance of wrinkles and expression lines and their deepening over time [24,25,26]. Injectable fillers have been used to correct age-induced connective tissue defects [24,25,26,27]. In our study, we constructed an EDC-crosslinked Ge/Ha hydrogel to investigate whether the skin’s histological and mechanical properties improve after its application.

Our hydrogels had a particle size inversely proportional to the EDC concentration used to crosslink Ge and Ha. This result is consistent with the degree of crosslinking between Ge and Ha observed at higher EDC concentrations [15]. Furthermore, when Ge and Ha are crosslinked with EDC at a high concentration in contrast to low concentrations, the hardness and solubility of hydrogels increase [15,28], which can directly affect their viscosity. Here, when the hydrogels had a larger particle size, the viscosity decreased. This may be because smaller particle sizes cause hydrogels to have higher resistance to flow, forming a more viscous structure.

As a result, EDC concentration directly affected the particle size and viscosity. This may be due to the many chemical bonds between Ge and Ha [28,29]. The EDC crosslinking mechanism depends on the reaction with the carboxyl groups of Ge and Ha, which can react with the amino groups of Ge, forming amide bonds [29,30]. In this sense, forming more bonds between Ge and Ha with EDC at 50 mM concentration decreased the viscosity of the gels. In addition, these hydrogels had a lower hydration capacity when analyzed with IR. It has been previously shown that Ge is more elastic at low crosslinking, while high crosslinking makes it stiff [31,32]. Furthermore, a high crosslinking of Ge improves the strength and glass transition temperature while decreasing the viscosity [31,33]. Essentially, the degree of crosslinking is related to all hydrogel characteristics.

The hydrophilic capacity of Ge and Ha is well known, but when they are crosslinked, this capacity is modified [32]. In our results, 50 mM EDC-crosslinked hydrogels presented lower hydration than 6 mM EDC-crosslinked hydrogels (Figure 4). Indeed, when the concentration of a crosslinker is increased, the hydrophilicity decreases due to the development of networks between the polymer chains [33,34]. Therefore, if a low crosslinker concentration is used, fewer bonds will be formed between the polymers, and a more significant number of free polymer chains will be present [34].

An ideal injectable gel should be highly biocompatible, easy to inject based on its favorable rheology, and should produce an acceptable long-lasting effect [34]. Similar to biocompatibility, the ability to mount a short inflammatory response and not a chronic reaction characterized by the formation of granulomas is essential [35]. In our results, applying all hydrogels resulted in an acute inflammatory response (Figure 5). Two days after the injection, an infiltrate of polymorphonuclear cells was observable. However, a gradual decrease in these cells was observed from that day on, and after day seven post-injection, they practically disappeared. Nevertheless, it is noteworthy that mononuclear cells, which corresponded to monocytes, peaked on day 4 post-application. After this, mononuclear cells continued decreasing until day 14, when they were no longer observed.

These results are consistent with an acute inflammation of short duration. A well-established element of acute inflammation is the recruitment and activation of neutrophils, which are rapidly attracted by chemoattractants [36,37] near the implantation site. The function of neutrophils is to destroy the biomaterial through phagocytosis, the secretion of proteolytic enzymes, and the generation of reactive oxygen species [38,39]. Once neutrophils begin to decrease, the cells that start to increase in the inflammatory infiltrate are monocytes that differentiate into macrophages once located at the implantation site [40]. There, they engulf foreign material and recruit other cell types, such as fibroblasts, which assist in tissue repair. Macrophages recognize the implant as a foreign object due to protein adsorption on the implant surface, resulting in macrophage differentiation and fusion into foreign body giant cells [41,42]. These cells are multinucleated with abundant cytoplasms and can reach a size of 100–150 µm [42,43]. The presence of macrophages and foreign body giant cells has been used as a marker of foreign body reaction [43]. On the other hand, monocytes can differentiate into two types of macrophages: (1) M1 macrophages, which secrete pro-inflammatory cytokines such as IL-1β, IL-6, IL-8, and TNFα [44]; and (2) M2 macrophages, which secrete anti-inflammatory cytokines, such as IL-10, and induce tissue remodeling [45]. Although we did not quantify M1 and M2 macrophages in our study, it could be that M1 macrophages were produced during the first phase to act against the hydrogel. At the same time, pro-inflammatory cytokines could contribute to the regeneration of the implant area. In the second phase, M2 macrophages could have been generated to cease the inflammatory process. However, it should be noted that the M1/M2 macrophage ratio after biomaterial implantation is essential for its immunological acceptance [40].

In our results, all hydrogels lasted a short time in the dermis after implantation. Figure 5 shows that the 3 mM and 6 mM EDC-crosslinked hydrogels were no longer present after seven days, and at 50 mM EDC, one implant disappeared between days 7 and 14 post-implantation. In contrast, the half-life of various commercial collagen/Ha hydrogels ranges from 2–3 months to 8–12 months [16]. This discrepancy may be due to the degree of crosslinking, as Ge/Ha hydrogels are more resistant to degradation when they are more crosslinked [32].

A higher degree of crosslinking in Ge/Ha scaffolds makes them more resistant to enzymatic degradation [31]. It is considered that the longer an injectable filler remains in the dermis, the better [46]. In the case of our gels, they only lasted a maximum of 14 days. However, this is not necessarily an adverse situation; if a scaffold lasts a short time in the connective tissue, there are possibly fewer adverse effects induced. Both Ge and Ha have been shown to cause adverse effects, including bruising, swelling, and excessive or insufficient volume, as well as infection, vascular occlusion, and blindness [46]. Histamine- and IgE-mediated type I hypersensitivity reactions and late T-cell-mediated type IV hypersensitivity reactions have also been described as complications following Ha injection [47,48,49].

It should be noted that despite the short permanence of the hydrogels in the dermis (Figure 5), an increase in fibroblasts was induced in the short-term and long-term effects (Figure 6). Concomitantly, an increase in the deposit of newly formed collagen was induced in both juvenile and old rats. It should be highlighted that these changes occurred even though the injected hydrogels had already disappeared. Non-crosslinked Ha has been observed to stimulate fibroblast activity in vitro, but not migration [50]. Non-crosslinked Ha is not used as a dermal filler due to its short half-life [27]. Both collagen/Ge and Ha play an essential role in developing and maintaining the extracellular matrix by allowing fibroblasts to adhere, proliferate, and differentiate [51]. The interaction of collagen/Ge and Ha with fibroblast surface receptors contributes to extracellular matrix remodeling, including synthesizing new collagen and Ha [52,53,54].

Collagen and Ge interact with fibroblasts through several integrins on the cell surface [54]. Collagen and Ge have binding sites for integrins with the β1 subunit and some with the alpha subunits α1, α2, α10, and α11 [55]. In addition, Ge-injected hydrogels induce the proliferation of keratinocytes, fibroblasts, and myofibroblasts [55]. Ha interacts with fibroblasts through several receptors, mainly the CD44 molecule [52]. This receptor is involved in cell adhesion, cell signaling, cytokine release, and extracellular matrix deposition [52,53,54,55,56,57]. Interestingly, Ge/Ha hydrogels induce the proliferation of fibroblasts and keratinocytes and the rapid closure of skin wounds [55]. In our results, an increase in fibroblasts was observed after applying the hydrogels, possibly because the fibroblasts were transiently attached to the hydrogels by integrins and CD44. Particularly, the increase in fibroblasts was more noticeable in old rats. This effect lasted three months but should be followed up for more long-term effects in future investigations.

Interestingly, in our study, an increase in the deposit of new collagen was observed after applying the hydrogels, both in juvenile and old rats. This has been previously observed, primarily with Ha. In a study by Wang, [58] forearm sun-damaged skin biopsies from human volunteers were analyzed after injection of crosslinked Ha, and an increase in collagen deposition was observed. In addition, skin that received crosslinked Ha injections showed increased collagen deposition around the fillers, an effect that was further increased for up to 13 weeks post-treatment. They found that the fibroblasts from the treated skin had a stretched morphology with a basophilic cytoplasm. Also, in vitro fibroblasts did not adhere to the hydrogel, indicating that the crosslinked Ha does not directly stimulate them; rather, it is the application of the gel that explains the mechanical stress [58]. Taken together with our results, it is suggested that the number of fibroblasts that proliferate and secrete an extracellular matrix may vary depending on the amount of injected hydrogel, the degradation time of the hydrogel in the tissue, and the degree of entanglement of the injected hydrogels. Further studies are needed to elucidate this effect.

Our results demonstrated a tendency of increased elastic moduli in juvenile and old rats after three months of treatment (Figure 7). Concerning the highest concentration of (50 mM) EDC used to crosslink the Ge/Ha hydrogels applied to juvenile rats, the elastic modulus value increased threefold compared with old rats. This result suggests that one subcutaneous application of Ge/Ha hydrogels in younger rats could offer more resistant skin for a long period. It is also possible that the elastic modulus increased due to the new collagen deposition (Figure 6c).

Mechanobiology studies the sensing capacity and cellular responses to mechanical microenvironments, such as the rigidity of the extracellular matrix or biomaterials. For example, it has been shown that extracellular matrices with different rigidities differentiate stem cells in cultures into different cell types [59]. Therefore, variations in the rigidity of the extracellular matrix in the skin may stimulate fibroblasts to behave differently, inducing them to divide and produce collagen or suppress their proliferation and decrease collagen secretion [60].

Several studies have highlighted the vital roles of mechanical signals in tissue regeneration, specifically in assisted skin repair [61]. There are two critical elements for the mechanical stability of the skin: the composition of the basement membrane and the extracellular matrix of the dermis [60]. Collagen IV, which forms a flexible three-dimensional network, is primarily responsible for providing rigidity to the basement membrane [62]. Decreased collagen IV in the basement membrane of embryonic skin induces mechanical stability [63]. However, the precise role of collagen IV in the mechanical strength of adult skin remains unclear. Here, we did not evaluate type IV collagen. Nevertheless, the changes in the dermis of the rats induced after subdermal injection of Ge/Ha hydrogels significantly increased skin tension.

It is well known that the dermal extracellular matrix comprises collagen I and III, which conjugate to form an intricate network providing mechanical resistance to the skin [64]. Notably, both collagen I and III are fibrillar, and the fact that they crosslink each other increases the rigid mechanical structure of the skin [65]. In our study, we observed a significant increase in newly synthesized collagen after the subdermal injection of Ge/Ha hydrogels, which may contribute to the increased mechanical stress of the skin, especially in the case of 50 mM EDC. In contrast, when the dermal collagen is fragmented, in addition to inducing aging, the mechanical tension of the skin decreases [64,65].

## 5. Conclusions

The subcutaneous application of the 50 mM Ge/Ha hydrogels effectively increased the number of fibroblasts, extracellular matrix, and skin tension in juvenile and old rats. Furthermore, an acute, low inflammatory response was observed after applying all hydrogels. The observed effects were similar in both juvenile and old rats. We propose the use of the Ge/Ha hydrogels as dermal fillers, highlighting the need to perform pre-clinical studies to evaluate other possible effects.

## Figures and Tables

**Figure 1 polymers-16-00573-f001:**
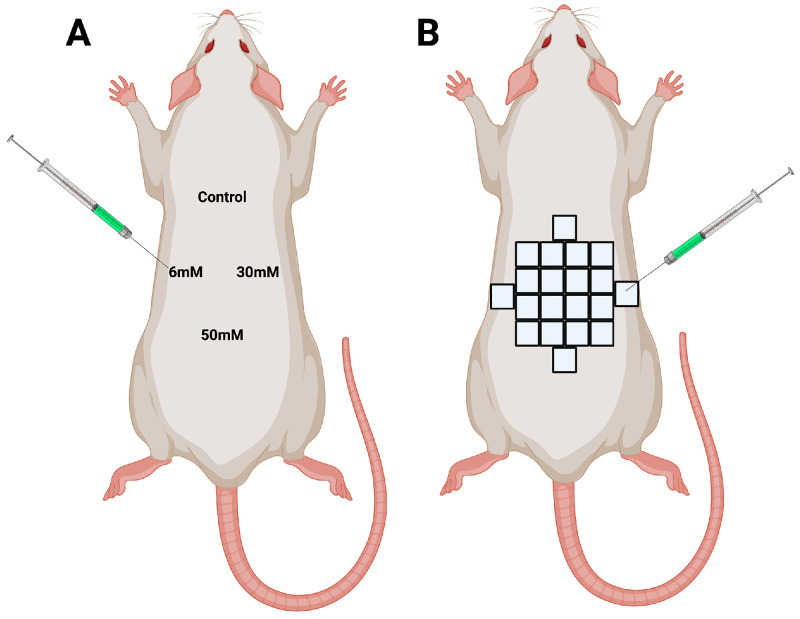
(**A**) Short-term treatment. Schematic representation of the application of hydrogels in rats (three months old) for evaluating hydrogel treatment with respect to short-term effects. (**B**) Long-term treatment. Schematic representation of the application of hydrogels in juvenile (three months old) and old rats (one-year-old) for the evaluation of the hydrogel treatment with respect to long-term effects.

**Figure 2 polymers-16-00573-f002:**
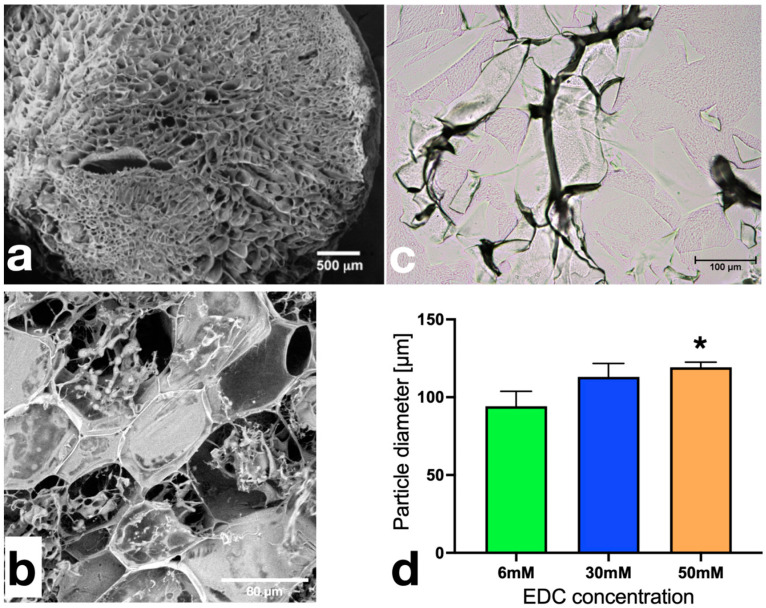
Structure of the Ge/Ha scaffolds and particle size of the Ge/Ha hydrogels. Structure of the porous 50 mM EDC-crosslinked Ge/Ha scaffold observed under SEM at (**a**) 100× and (**b**) 1000× magnification. (**c**) A representative microphotograph of a Ge/Ha scaffold after pulverization. The observed particles are presented as irregular trabeculae (1500×). (**d**) Analysis of the particle size after Ge/HA scaffold pulverization. The size of the trabeculae augmented alongside ascending concentrations of EDC. * *p* < 0.0001.

**Figure 3 polymers-16-00573-f003:**
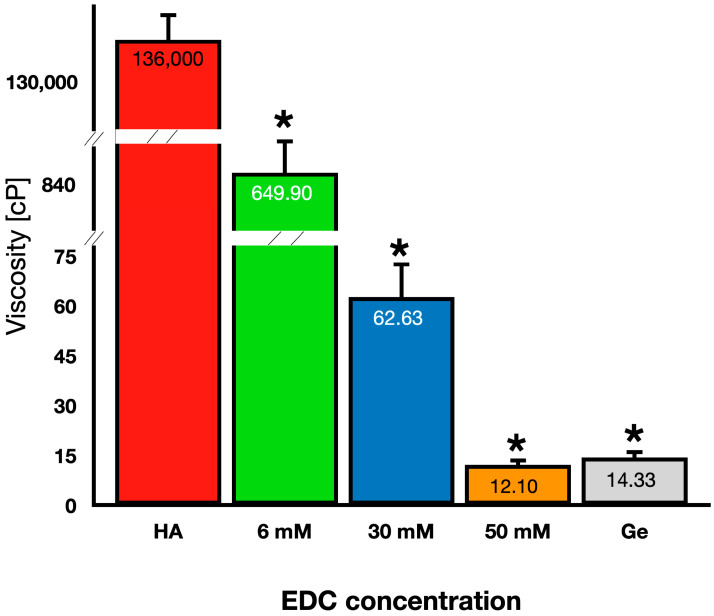
Quantification of the viscosity of EDC-crosslinked Ge/Ha hydrogels. The viscosity of the hydrogels significantly decreased when increasing concentrations of EDC were used for the crosslinking of the Ge/Ha scaffolds. * *p* < 0.0001.

**Figure 4 polymers-16-00573-f004:**
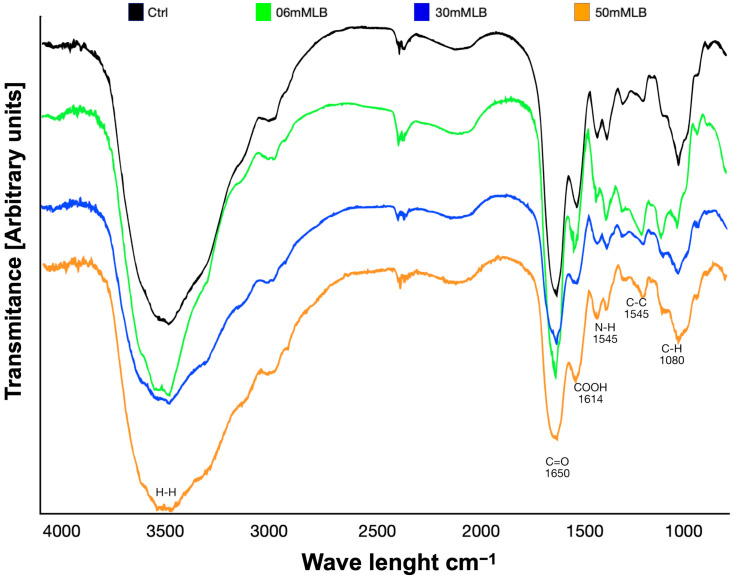
IR analysis of the composition of the Ge/Ha hydrogels. The presence of active groups was observed at the same wavelength as the control (without crosslinking) and EDC-crosslinked hydrogels. The same peaks can be observed in all curves, but the hydration level is inversely proportional to the concentration of EDC.

**Figure 5 polymers-16-00573-f005:**
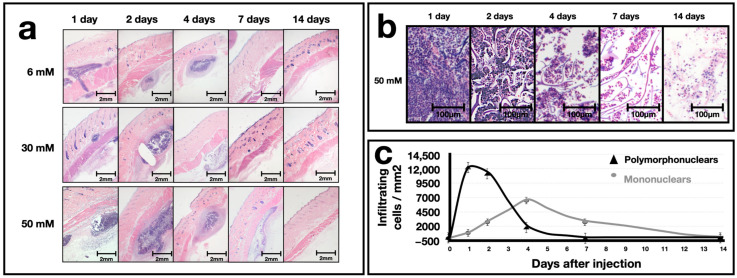
Evaluation of the inflammatory response induced by the subcutaneous administration of the Ge/HA hydrogels. (**a**) Holoptic images of a rat at low magnification (20×), depicting the dorsal skin after punctual administration of 50 µL of Ge/Ha hydrogels. In the case of 6 mM and 30 mM EDC, infiltrating inflammatory cells can be observed in the dermis from day one until day four post-application. In the case of 50 mM EDC, infiltrating inflammatory cells are observed until day seven post-application. In general, inflammatory cells were observable on day seven post-application and gradually decreased until they disappeared on day fourteen. (**b**) Representative microphotographs of the infiltrating inflammatory cells induced by applying the hydrogels in the dermis. At one and two days post-application, many leukocytes, mainly polymorphonuclear cells, were present around the particles that form the hydrogel. The number of inflammatory cells decreased on day four post-application. Finally, on day 14, the inflammatory infiltrate and particles that form the hydrogel decreased considerably. (**c**) Quantification of leukocytes at the application site. During the first days after treatment, the polymorphonuclear cells were the most abundant cells. However, on days three and four, polynuclear cells began to decrease and mononuclear cells increased.

**Figure 6 polymers-16-00573-f006:**
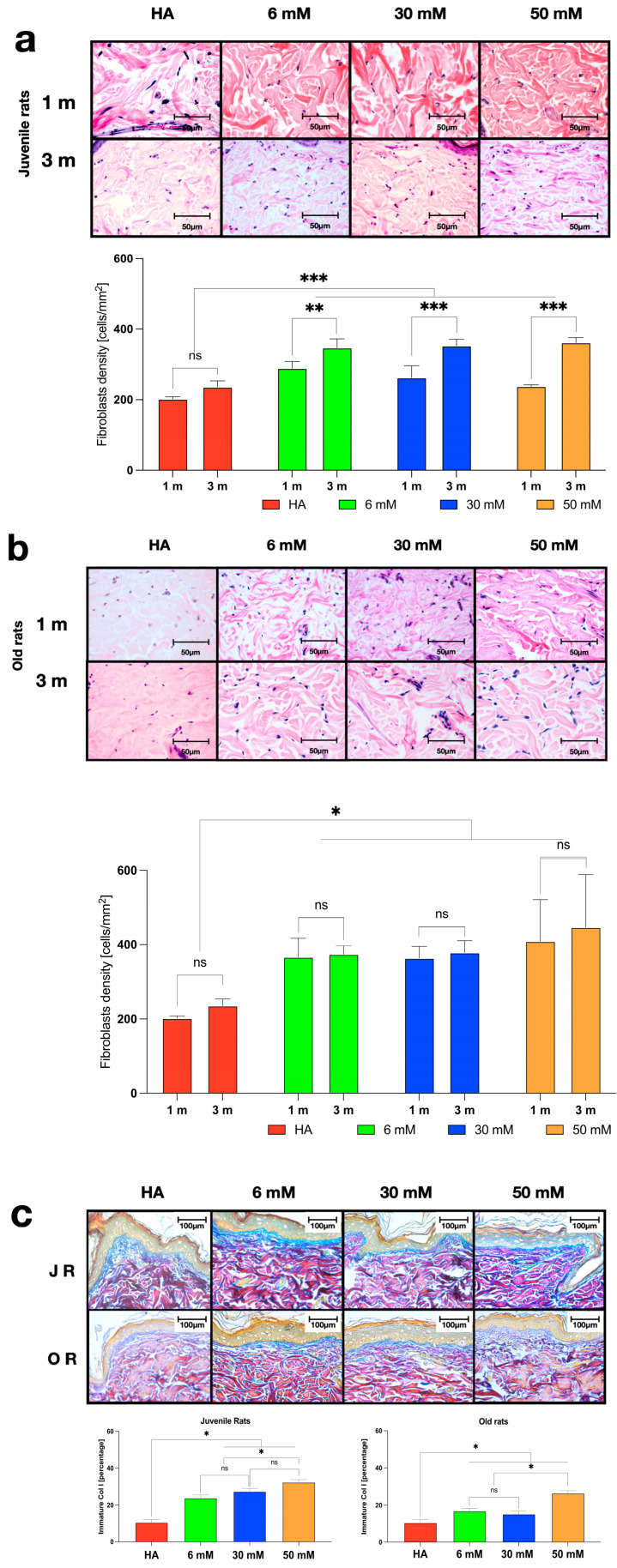
Photomicrographs of skin samples of juvenile and old rats after the application of Ge/Ha hydrogels. Representative images of the reticular dermis (H&E, 400×) after one and three months of treatment in the groups of (**a**) juvenile and (**b**) old rats. Quantification of the number of fibroblasts per square millimeter for the groups of (**a**) juvenile and (**b**) old rats (** *p* = 0.001; *** *p* = 0.0002, * *p* = 0.0001; ns: non-significant changes). (**c**) Representative images of the whole dermis of rats at three months post-application of the hydrogels. Herovici staining (400×). In the photomicrographs, the epidermis is shown in green, while the papillary and reticular dermis is shown in cerulean blue and red, depending on their maturation degree. The region stained in cerulean blue allows for the identification of the recently synthesized extracellular matrix formed by type I collagen (* *p* = 0.0002).

**Figure 7 polymers-16-00573-f007:**
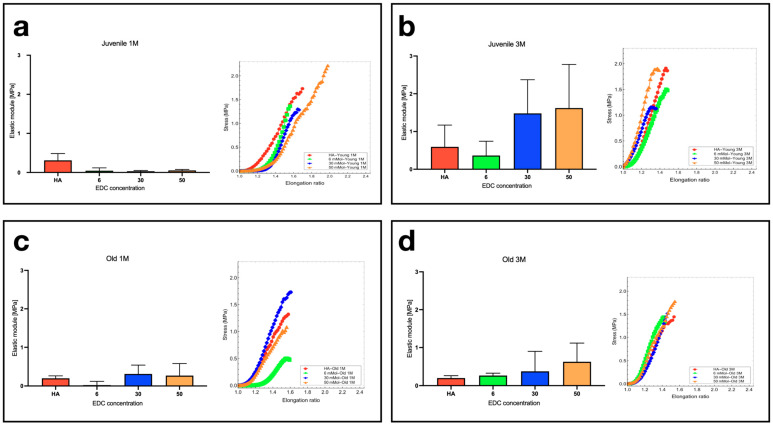
Average values of the elastic moduli of rats’ skin in a longitudinal direction after one and three months of treatment with EDC-crosslinked Ge/Ha hydrogels. The right-side charts of each panel depict the non-linear mechanical response of the rats’ skin. (**a**) Longitudinal elastic modulus in juvenile rats after one month and (**b**) three months of treatment. (**c**) Longitudinal elastic modulus in old rats after one month and (**d**) three months of treatment.

**Table 1 polymers-16-00573-t001:** Experimental conditions. Temporal determination, experimental groups, number of hydrogel inoculations and biopsies obtained.

Experiments	Time	Conditions	Injections	Biopsies
Short term	1, 2, 4, 7, and 14 days	Control, 6, 30, and 50 mM	50 μL of each condition were inoculated in one rat. Each rat received 4 inoculations (Figure 1A) Three rats per time.	4 biopsies (one for each condition) in one rat. 3 rats per time. 15 rats in total with 60 biopsies.
Long term	1 and 3 months	Control, 6, 30, and 50 mM	20 inoculations of 50 μL in 3 rats per condition, distributed in the dorsal skin (Figure 1B)	A biopsy from a place treated with 50 μL of each condition in one rat. The rest of the skin was used for mechanical tests. 3 rats per condition (12 biopsies total in each experimental time).

## Data Availability

The data presented in this study are available on request from the corresponding author.

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
