# Peer review of "Subcutaneous Application of a Gelatin/Hyaluronic Acid Hydrogel Induces the Production of Skin Extracellular Matrix"

_polymers, 2024, doi:10.3390/polym16050573_

Round 1

Reviewer 1 Report

Comments and Suggestions for Authors

The manuscript entitled “Subcutaneous application of a gelatin-hyaluronic acid hydrogel 2 induces the production of skin extracellular matrix” describes the use of EDC-crosslinked gelatin-hyaluronic acid (Ge-Ha) hydrogels as dermal filler materials in a rat model. Since a similar kind of Ge-Ha gel material has been used recently for soft tissue regeneration (for example https://www.ecmjournal.org/papers/vol024/pdf/v024a23.pdf), this application in dermal fillers is well reasoned, though not extremely novel. The authors conclude that their hydrogel stimulates collagen production in the dermis and stiffens skin, though the implants are short lived at only 1-2 weeks, compared to many months for most commercial dermal fillers, and the acute inflammatory response is reasonably low and well tolerated by the rodents. These results may motivate further preclinical studies on the efficacy of the material in dermal filler treatments

It appears that the authors have carried out their study in a scientific and ethical manner, and the manuscript is generally well-written with minimal grammatical or syntax problems. There is a large amount of data, and the discussion is well-grounded in literature citations. As such, the manuscript holds promise for publication. However, a number of major concerns should be addressed by the authors before publication of the manuscript.

The methods are unclear. Section 2.5 says 40 rats were involved in the study, with N=3 per condition, but it is not clear what all the conditions are. Are the conditions 1 2 4 7 and 14 days? Are the conditions young and old rats with the same 50 ul implant? What about the long-term group receiving 20 injections? It is strongly suggested to include a table to explain more clearly how many treatment groups and conditions there are, accounting for all rates involved in the study.

Further clarification is also needed for histology. How many rats per treatment group were biopsied?

The authors need to clarify how they performed statistical analysis of treatment groups. With only 3 individuals per condition and 40 rats, in theory there are 40/3 = 13.3 treatment groups. With these very low group sizes and large number of groups, statistical significance would be almost impossible to observe, especially when applying appropriate multiple-comparison confections such as the Bonferroni test or other similar corrections.

Figures 1 and 2 can be merged into a single figure with parts (a) and (b), and the text on the rat illustrated in Figure 1 needs to be enlarged. It is not legible at the current size and resolution. The captions of these figures should further clarify the illustrations and the marks on the back; currently the captions and figures do not make sense to this reviewer. For example, the captions should explain the location(s), number, and volume of injections.  

Figures 7 and 8 are not of publication quality. The bar graphs in Figure 7 are far too small to read – the test is illegible. The inset graphs in Figure 8 are also far too small to read the numbers and axis labels.

The data in Figure 8 appears to be collected from a single sample and single test. Uniaxial tests may have considerably large scatter in the data sample to sample, so the data cannot be trusted without multiple attempts and error bars. It is suggested that these data be eliminated from the manuscript so as not to mislead readers who don’t notice that N=1. If these tests were repeated multiple times, then the uncertainty of the measurements should be specified and illustrated in the data.

Reviewer 2 Report

Comments and Suggestions for Authors

Comment to Author

I read the paper entitled " Subcutaneous application of a gelatin-hyaluronic acid hydrogel induces the production of skin extracellular matrix" done by a Jarquín-Yáñez et al. The manuscript looks interesting and this is a worthwhile subject. However, some major and minor concerns exist regarding this paper as following. The following point should be added to the revised manuscript and changed.

Please modify this in the entire manuscript

Can you tell me about the originality of this paper? Many papers concerning the HA-Ge are crosslinked by either the EDC or the EDC/NHS. Your originality and novelty should be boldly expressed.

1.       Introduction section.

o   Page 2, line 52 " Hydrogels are chemically stable, crosslinked hydrophilic polymer chains with multiple biomedical and pharmaceutical applications as dermal fillers”. I believe that the sentence is unclear and incorrect. There are numerous hydrogels that are not stable.

o   Page 2 line 72-76, "N-(3-Dimethylaminopropyl)-N´- ethylcarbodiimide (EDC) is a chemical crosslinker that creates covalent bonds between the hydroxyl and amino groups of the compounds, leading to the formation of amides and esters that functionalize the Ha hydrogel linked to another natural polymer, such as Ge". I believe that this is incorrect or that it needs to be modified. The amide is formed by the EDC between the amine and carboxylic group (which is a more stable bond than an ester). It is necessary for the author to rewrite and modify it in order to avoid giving the reader of the journal inaccurate information.

2. materials and methods section

o   You should include a section on the statistical analysis in your study and which test you used (t-test and what is considered statistically significant)

3. Result and discussion;

o   The quality of figure 6 is so poor. I suggest the author arrange the material so that we can see the big picture as well as the details. When looking at the histology, it is difficult to determine what occurred.

o   The author is advised to check the in vitro biodegradability after 14 days to compare it with the in vivo degradation.

Round 2

Reviewer 2 Report

Comments and Suggestions for Authors

Since the author addressed the comment well, I believe that this version of the paper can be published in a journal.